# Exploring Potential Antimalarial Candidate from Medicinal Plants of Kheaw Hom Remedy

**DOI:** 10.3390/tropicalmed7110368

**Published:** 2022-11-10

**Authors:** Prapaporn Chaniad, Tachpon Techarang, Arisara Phuwajaroanpong, Natharinee Horata, Abdi Wira Septama, Chuchard Punsawad

**Affiliations:** 1Department of Medical Sciences, School of Medicine, Walailak University, Nakhon Si Thammarat 80160, Thailand; 2Research Center in Tropical Pathobiology, Walailak University, Nakhon Si Thammarat 80160, Thailand; 3Faculty of Medical Technology, Huachiew Chalermprakiet University, Bang Phli 10540, Thailand; 4Research Center for Pharmaceutical Ingredient and Traditional Medicine, National Research and Innovation Agency (BRIN), Cibinong Science Center, Bogor 16915, Indonesia

**Keywords:** antimalarial activity, acute toxicity, *Globba malaccensis*, *Plasmodium berghei*, *Plasmodium falciparum*, Kheaw Hom remedy

## Abstract

The Kheaw Hom remedy is a traditional Thai medicine widely used to treat fevers. Some plant ingredients in this remedy have been investigated for their antimicrobial, antiviral, anti-inflammatory, and antioxidant activities. However, there have been no reports on the antimalarial activities of the medicinal plants in this remedy. Therefore, this study focuses on identifying potential antimalarial drug candidates from the medicinal plant ingredients of the Kheaw Hom remedy. Eighteen plants from the Kheaw Hom remedy were extracted using distilled water and ethanol. All extracts were investigated for their in vitro antimalarial activity and cytotoxicity. An extract that exhibited good in vitro antimalarial activity and low toxicity was selected for further investigation by using Peter’s 4-day suppressive test and an acute oral toxicity evaluation in mice. Based on the in vitro antimalarial activity and cytotoxicity studies, the ethanolic extract of *Globba malaccensis* rhizomes showed promising antimalarial activity against the *Plasmodium falciparum* K1 strain (IC_50_ = 1.50 µg/mL) with less toxicity to Vero cells (CC_50_ of >80 µg/mL). This extract exhibited a significant dose-dependent reduction in parasitemia in *P. berghei*-infected mice. The maximum suppressive effect of this extract (60.53%) was observed at the highest dose administered (600 mg/kg). In a single-dose acute toxicity test, the animals treated at 2000 mg/kg died within 48 h after extract administration. In conclusion, our study indicates that the ethanolic extract of *G. malaccensis* rhizomes exhibited in vitro and in vivo antimalarial activities, which could serve as a promising starting point for antimalarial drug.

## 1. Introduction

Malaria is an infectious disease caused by protozoan parasites of the genus *Plasmodium* that is transmitted to humans by the bite of an infected female *Anopheles* mosquito [1,2]. Parasites reproducing in the erythrocytic cycle can induce infected red blood cells to rupture, leading to a high body temperature increase in malaria patients [1]. There are an estimated 228 million cases of *Plasmodium* infection worldwide and 405,000 malarial deaths, most of which are found in two high-risk groups, pregnant women and children [2]. There were 11 million pregnant women infected with malaria, and children under 5 years of age accounted for 67% of all malaria deaths in sub-Saharan Africa [2]. The clinical symptoms of malaria are divided into three stages: the cold stage (shivering), the hot stage (fever and/or headaches), and the sweating stage (sweats) [1,2]. However, severe malaria occurs in complicated infections or due to organ failure or abnormalities of metabolism after malaria infection, increasing the morbidity and mortality rate in these patients. There are several malarial complications, such as cerebral malaria, severe anemia, acute lung injury/acute respiratory distress syndrome, acute kidney injury, metabolic acidosis, and hypoglycemia [1]. To date, there is drug resistance to *Plasmodium* parasites in most malaria-endemic countries, especially the region of Southeast Asia, such as the border areas of Thailand, Cambodia, and Myanmar, which have the highest risk for multiple drug-resistant infections [3]. Resistance has been described for two of the five *Plasmodium* species that cause malaria infection in humans, including *P. falciparum* and *P. vivax* [3]. Therefore, we need a new drug to inhibit *Plasmodium* parasites by using traditional Thai remedies.

The Kheaw Hom remedy is a Thai traditional medicine that was published in the National List of Essential Medicines 2011 [4]. Folk doctors have long been using this remedy to treat fever in high-fever groups, such as patients with chickenpox, measles, and herpes zoster [4]. Kheaw Hom remedy consists of eighteen Thai medicinal plants with cooling and bitter characteristics that decrease toxins in the blood. Kheaw Hom remedy is recommended for both ingestion and application of the medicine on the skin [4]. Previous studies have reported that the Kheaw Hom remedy has antimicrobial, antiviral, anti-inflammatory, and antioxidant activities [5,6,7,8,9]. One of the plant materials in the Kheaw Hom remedy is the previously described *Globba malaccensis* [4,6]. *G. malaccensis* a perennial plant that grows to a height of 1 m. Its leaves are green with lanceolate, long petioles that taper at each end and are smooth, 10–15 cm wide, and 30–35 cm long. The roots have an aromatic and hot fragrance [6]. Folk doctors have used the rhizomes of *G. malaccensis* to treat diarrhea and centipede, snake, or scorpion bites [6]. Previous studies have reported that the ethanolic extract of *G. malaccensis* rhizomes has high anti-inflammatory activity through the inhibition of nitric oxide (NO) production and free radical scavenging activity [10,11]. However, there are no research reports that verify its activity against infectious diseases that cause high fever, such as malaria.

The Kheaw Hom remedy is widely used by Thai folk doctors against groups of diseases that cause high fevers. However, there have been no reports of its antimalarial activities. Therefore, the objective of this investigation was to study the antimalarial activities of the Kheaw Hom remedy and its plant ingredients.

## 2. Materials and Methods

### 2.1. Plant Collection and Management

The plant materials in the Kheaw Hom remedy were purchased from a traditional Thai drug store in Nakhon Si Thammarat Province, Thailand. These materials consist of eighteen traditional plants, namely *Pogostemon cablin* (Blanco) Benth., *Limnophila rugosa* (Roth) Merr., *Cordyline fruticosa* (L.) A.Chev. (Red leaves), *Cordyline fruticosa* (L.) A.Chev. (Green leaves), *Eupatorium stoechadosmum* Hance., *Vetiveria zizanioides* (L.) Nash ex Small, *Kaempferia galanga* L., *Tarenna hoaensis* Pit., *Dracaena loureiroi* Gagnep., *Angiopteris evecta* (G. Forst.) Hoffm., *Globba malaccensis* Ridl., *Tacca chantrieri* André., *Sophora exigua* Craib., *Cyathea podophylla* (Hook.) Copel., *Mimusops elengi* L., *Mesua ferrea* L., *Mammea siamensis* Kosterm., and *Nelumbo nucifera* Gaertn. (Table 1). The Kheaw Hom remedy was purchased from the company Vejpongosot, Bangkok, Thailand. The identification of all materials was performed using botanical characteristic patterns by Assoc. Prof. Tanomjit Supavita, School of Pharmacy, Walailak University. Prior to extraction, the plant materials were washed with distilled water to remove dust and dead matter. Then, the plants were dried in a hot air oven at 60 °C, coarsely chopped in an herb grinder, and kept separate from each other in a closed cabinet until use.

### 2.2. Plant Extraction

The powdered plant samples were extracted by two different extraction methods, heat reflux and maceration, by using distilled water and ethanol, respectively [5,12]. To increase the solubility and diffusion of solutes, the heat reflux method using water as a solvent was selected to extract the sample. Briefly, 600 mL of distilled water was added to 60 g of the powdered plant samples in a round bottom flask. Then, the mixture was heated and refluxed for 2 h. In another set of experiments, the powdered plant samples (60 g) were blended with 600 mL of 95% ethanol and macerated for 3 days at room temperature. The residue from each method was extracted again using fresh solvent under the same conditions mentioned above. Each solvent extraction was filtered through gauze and Whatman No. 1 filter paper. The filtered extracts were dried with a rotary evaporator (Buchi, Shanghai, China). Finally, the dried plant extracts were kept in screw cap containers and stored at 4 °C for further use.

### 2.3. Phytochemical Analysis

Qualitative phytochemical screening analysis was conducted on the eighteen extracted plants from the Kheaw Hom remedy to identify the presence of secondary metabolites in each of the ethanolic and aqueous extracts. These secondary metabolites included flavonoids, terpenoids, alkaloids, tannins, anthraquinones, cardiac glycosides, saponins, and coumarins. The methods used for each phytochemical constituent were performed according to previous studies [5,13].

### 2.4. In Vitro Cultivation and Maintenance of Plasmodium falciparum

Cultivation of the *P. falciparum* K1 strain was performed according to a previous study with some modifications [14]. Briefly, the *P. falciparum* K1 strain was cultured and maintained in 5 mL of culture medium (RPMI-1640) with 2 mg/mL sodium bicarbonate, 10 μg/mL hypoxanthine (Sigma-Aldrich, New Delhi, India), 4.8 mg/mL HEPES (HiMedia, Mumbai, India), 0.5% Albumax II (Gibco, Waltham, MA, USA), and 2.5 μg/mL gentamicin (Sigma-Aldrich, New Delhi, India) at a 3% hematocrit suspension in human red blood cells. The cultures were incubated at 37 °C in a CO_2_ incubator. The percentage of parasitemia was monitored daily, and the percentage of noninfected type O-positive red blood cells was maintained.

### 2.5. In Vitro Antimalarial Activity Assay

Both the ethanolic and aqueous extracts of each plant were tested for their antimalarial activity by using an in vitro parasite lactate dehydrogenase (pLDH) assay [15]. Briefly, a 2% parasitemia parasite suspension with 2% hematocrit was added to a 96-well culture plate. Then, 1 µL of crude extract at final concentrations in the range of 0.3 to 2500 mg/mL was added to each well. Artesunate at concentrations in the range of 0.3–10.0 µg/mL (Sigma-Aldrich, New Delhi, India) and dimethyl sulfoxide (DMSO) (Merck, Darmstadt, Germany) were also added to wells as positive and negative controls, respectively. Noninfected red blood cells were used as a blank control. The plate was incubated at 37 °C with 5% CO_2_ for 72 h in a CO_2_ incubator, frozen three times at 20 °C, and thawed at 37 °C. Next, 20 µL of each sample was removed and transferred to a new microplate, which contained Malstat reagent (Sigma-Aldrich, New Delhi, India). Nitroblue tetrazolium/phenazine ethosulfate solution (Sigma-Aldrich, New Delhi, India) was added to each well of the plate and stored in the dark for 60 min. To stop the reaction, 5% acetic acid (Merch, Darmstadt, Germany) was added to each well. The absorbance of each well was then measured at 650 nm by using a microplate reader. Each assay was performed in triplicate. Finally, the percent inhibition and half maximal inhibitory concentration (IC_50_) were calculated by using a nonlinear dose–response curve.

### 2.6. In Vitro Assessments of Cytotoxicity

Ethanolic and aqueous extracts of eighteen medicinal plants and the Kheaw Hom remedy were tested for their cytotoxicity by using a 3-(4,5-dimethylthiazol-2-yl)-2,5-diphenyltetrazolium bromide (MTT) assay Kit (Cell Proliferation, cat. no. ab211091, Abcam, UK). This colorimetric assay is based on the capacity of mitochondrial succinate dehydrogenase enzymes in living cells to reduce the yellow water-soluble substrate MTT into an insoluble, colored formazan product, the absorbance of which can be measured spectrophotometrically [16]. Briefly, African green monkey kidney normal cells (Vero cells) were added to a 96-well culture plate at a density of 1 × 10^4^ cells/mL and incubated in complete medium containing 10% fetal bovine serum at 37 °C with 5% CO_2_ for 24 h. After monolayer formation, the supernatant was discarded, and 100 µL of each crude extract at different concentrations, ranging from 5 to 80 µg/mL, were added to the culture plate and incubated at 37 °C in 5% CO_2_ for 48 h. DMSO and a twofold dilution of doxorubicin were used as negative and positive controls, respectively. Untreated cells were also used as a control. After incubation, the sample solution was changed, and 100 µL of fresh media containing the MTT reagent was added to each well. Then, the culture plates were incubated at 37 °C in 5% CO_2_ for 3 h, followed by the addition of 50 µL of propanol, wrapping, and gentle shaking to solubilize the formed formazan. Finally, the absorbance was measured at 590 nm by using a microplate reader. The 50% cytotoxic concentration (CC_50_) was calculated from the dose–response curve.

### 2.7. Four-Day Suppressive Test (Peter’s Test)

This test was used to measure the ability of the ethanolic and aqueous extracts against the schizont stage of *Plasmodium berghei*-infected ICR mice. The 4-day suppressive test was performed according to a previous study with some modifications [12,17]. Male ICR mice were randomly divided into five groups of 5 mice each, as shown in Table 2. All mice were injected with 1 × 10^7^ red blood cells infected with *P. berghei* ANKA via intraperitoneal injection [12]. Treatment started 4 h after the mice had been inoculated with the *Plasmodium* parasite. The negative control group received 200 µL of 7% Tween 80 solution, whereas the positive control group was given 6 mg/kg body weight artesunate orally per day. For each extract treatment group, the animals received daily oral doses of 200, 400, or 600 mg/kg body weight ethanolic extract of *G. malaccensis* as low, moderate, and high doses, respectively, according to previous studies [12,18]. The mice were given each substance daily for 4 days (at 24, 48, and 72 h after induction). The body weights of all mice were measured on days 1 and 5 of the experiment. On day 5, parasitemia was investigated using the Giemsa staining technique. At the end of the experiment, the mice were euthanized by injection of 200 mg/kg body weight pentobarbital. Finally, the percent inhibition was calculated using the following equation.
(1)$$\%\;\mathrm{inhibition}=\frac{\left(\mathrm{parasitemia}\;\mathrm{of}\;\mathrm{negative}\;\mathrm{group}\;-\;\mathrm{parasitemia}\;\mathrm{of}\;\mathrm{treated}\;\mathrm{group}\right)}{\left(\mathrm{parasitemia}\;\mathrm{of}\;\mathrm{negative}\;\mathrm{group}\right)}\;\times \;100$$

### 2.8. Acute Toxicity Test

The crude ethanolic extract of *G. malaccensis* was assessed for its toxicity in noninfected ICR mice aged 6–8 weeks weighing 25–35 g according to the standard guidelines of the Organization for Economic Cooperation and Development (OECD) [19]. The mice were randomly divided into three groups: 2000 mg/kg body weight *G. malaccensis* extract-treated, 7% Tween 80-treated, and untreated groups. Before the experiment, all mice were fasted for 3 h with access to only drinking water. The ethanolic extract of *G. malaccensis* was dissolved in 7% Tween 80 to a concentration of 2000 mg/kg body weight. In the treatment group, the mice orally received a single dose of 2000 mg/kg body weight *G. malaccensis* extract, while the control group was orally administered 200 µL of a 7% Tween 80 solution. Physical and behavioral changes in the mice were observed 14 days after administration, including rigidity, sleep, diarrhea, depression, abnormal secretion, hair erection, mortality, and other manifestations of toxicity. At the end of the experiment, all mice were injected with 60 mg/kg body weight pentobarbital. The cardiac puncture technique was used to collect blood samples into heparinized tubes for biochemical analysis. Additionally, the liver and kidney tissues were removed for histopathological examination.

### 2.9. Selectivity Index

To estimate the potential of the extracts to inhibit the growth of parasites without toxicity, the selectivity index (SI) was calculated using the following equation.
Selectivity index (SI) = CC_50_ Vero cells/IC_50_
*P. falciparum*(2)

### 2.10. Ethical Statement

The study protocol was approved by the Human Ethics Committee of Walailak University, Thailand (approval number: WUEC-19-122-01). The approved protocols for the animal study were obtained from the Animal Ethics Committee of Walailak University, Thailand (protocol number: WU-AICUC-63001). All protocols in this study were carried out in accordance with the relevant guidelines and regulations for using animals in compliance with Animal Research: Reporting of In Vivo Experiments (ARRIVE) guidelines.

### 2.11. Statistical Analysis

The results are presented as the means ± SEM. Statistical analysis was performed using IBM SPSS Statistics version 23.0 software (IBM, Armonk, NY, USA). The Kolmogorov–Smirnov goodness-of-fit test was used to test normal distribution. The statistical significance of parasitemia inhibition was analyzed by using one-way analysis of variance (ANOVA), followed by Tukey’s multiple comparison test. The level of significant difference was set at a *p*-value less than 0.05 (*p* < 0.05).

## 3. Results

### 3.1. Phytochemical Screening

Phytochemical screening of the extracted plants from the Kheaw Hom remedy is summarized in Table 3. The extracted plants from the Kheaw Hom remedy showed that flavonoids, tannins, terpenoids, saponins, alkaloids, coumarins, and anthraquinones were present. However, cardiac glycoside was not observed in any of the ethanolic or aqueous extracts of the plants from the Kheaw Hom remedy.

### 3.2. In Vitro Antimalarial Activity and Cytotoxicity of the Plant Extracts from Kheaw Hom Remedy

The results of the in vitro antimalarial activity against the *P. falciparum* K1 strain are presented in Table 4, which shows the concentration of each extract that inhibited 50% of parasitic growth (IC_50_). It should be noted that a lower IC_50_ value indicates a stronger activity against *P. falciparum*. In this study, the antimalarial activities of all extracts were classified based on their IC_50_ values as follows: very good activity (IC_50_ < 1 μg/mL); good activity (IC_50_ 1–10 μg/mL); moderate activity (IC_50_ > 10–50 μg/mL); weak activity (IC_50_ > 50–100 μg/mL); inactive (IC_50_, >100 μg/mL). Of the thirty-eight extracts of Kheaw Hom and its plant ingredients, the extracts from three plants possessed good antimalarial activity. The ethanolic extract of *M. siamensis* possessed the highest activity, with an IC_50_ value of 1.50 µg/mL, followed by the ethanolic extract of *G. malaccensis* and the ethanolic extract of *M. ferrea*, with IC_50_ values of 5.33 and 7.39 µg/mL, respectively. The extracts from five plants, i.e., the ethanolic extracts of *D. loureiroi*, *P. cablin*, *K. galangal*, *E. stoechadosmum*, and *M. elengi*, exhibited moderate activity with IC_50_ values of 10.47, 24.49, 25.80, 28.15, and 49.80 µg/mL, respectively. The Kheaw Hom remedy also exhibited moderate activity with an IC_50_ value of 14.16 µg/mL. Notably, the ethanolic extracts of almost all plants except for *N. nucifera* and *S. exigua* exhibited antimalarial activity greater than that of the aqueous extracts. Two of these extracts presented different phytochemical constituents; flavonoids and terpenoids were predominantly present in the ethanolic extracts to a greater extent than in the aqueous extracts, leading to the ethanolic extracts showing more potent antimalarial effects.

Most of the aqueous extracts were inactive (IC_50_ > 100 µg/mL), except for four extracts that exhibited weak activity, i.e., *N. nucifera*, *T. chantrieri*, *M. siamensis*, and *L. rugose* (IC_50_ = 53.08–83.76 µg/mL). The cytotoxic effects on Vero cells of all plant extracts from the Kheaw Hom remedy are displayed in Table 4. All plant extracts and the Kheaw Hom remedy were nontoxic to Vero cells, with CC_50_ values greater than 80 µg/mL, except for the aqueous extracts of *M. siamensis*, *M. ferrea*, *S. exigua*, and *D. loureiroi*, which revealed toxic effects, with CC_50_ values of <5, 11.44, 18.56, and 55.67 µg/mL, respectively.

From these results, promising antiplasmodial activity against the *P. falciparum* K1 strain was found in the ethanolic extracts from three plants: *M. siamensis* flowers (IC_50_ = 1.50 µg/mL), *G. malaccensis* rhizomes (IC_50_ = 5.33 µg/mL), and *M. ferrea* flowers (IC_50_ = 7.39 µg/mL). Additionally, the ethanolic extract of *M. siamensis,* which displayed the most potent antimalarial activity, showed a toxic effect on Vero cells (CC_50_ < 5 µg/mL), while the ethanolic extract of *G. malaccensis* was nontoxic to Vero cells with a CC_50_ of >80 µg/mL; the ethanolic extract of *M. ferrea* also showed a toxic effect on Vero cells (CC_50_ = 11.44 µg/mL) (Table 4). The SI values of *M. siamensis* flowers, *G. malaccensis* rhizomes, and *M. ferrea* flowers were <3.33, >15.01, and 1.55 against *P. falciparum*, respectively (Table 4). Therefore, the ethanolic extract of *G. malaccensis*, with promising antimalarial activity and low Vero cell toxicity, was further selected for examination in the 4-day suppressive test and acute toxicity test in mice.

### 3.3. Four-Day Suppressive Test

The results of this study showed that the mice treated with the ethanolic extract of *G. malaccensis* displayed a significant decrease in parasite count when compared to the negative control group. In addition, the *G. malaccensis* ethanolic extract at all treated doses significantly suppressed the *Plasmodium* parasite in a dose-dependent manner. However, the parasite was not cleared completely from any of the *G. malaccensis*-treated groups, whereas the positive control group treated with artesunate, used as a standard antimalarial drug, at a daily dose of 6 mg/kg body weight, cleared more than 90% of the parasites (Figure 1, Table 5). Furthermore, the extract-treated groups also had a change in body weight, as shown in Table 6. The body weights of all animals in the *G. malaccensis* extract-treated groups significantly decreased compared with the negative and positive control groups.

### 3.4. Acute Toxicity Test

According to the acute oral toxicity study of ethanolic extract of *G. malaccensis*, 66.67% of the mice died within the first 48 h after receiving a single dose of 2000 mg/kg, and all of the mice died within 72 h. Moreover, signs of toxicity were noted from the physical and behavioral observations of the treated mice, such as decreased motor activity, decreased response to touch, decreased appetite, and hair erection, in all mice after first receiving the first dose of *G. malaccensis* extract. Based on these results, the 50% oral lethal dose (LD_50_) of *G. malaccensis* extract is less than 2000 mg/kg.

## 4. Discussion

The Kheaw Hom remedy has been used in Thai traditional medicine to reduce high fever in patients [4]. The aqueous and ethanolic extracts of all plants from the Kheaw Hom remedy were screened for the presence of phytochemical constituents, and all of the extracts were then tested in an in vitro assay for their antimalarial activity against the K1 strain of *P. falciparum* and their cytotoxicity to Vero cells. Phytochemical screening demonstrated that the extracts of the plants of the Kheaw Hom remedy contain flavonoids, terpenoids, alkaloids, tannins, anthraquinones, saponins, and coumarins. However, cardiac glycosides were not present in any of the plants of the Kheaw Hom remedy. In this study, the flavonoids, terpenoids, and alkaloids found were mostly present in the Kheaw Hom remedy ethanolic extracts. In the in vitro antimalarial studies, highly promising starting points for antiplasmodial activity were found in three ethanolic extracts: *M. siamensis*, *G. malaccensis*, and *M. ferrea*. Based on the cytotoxicity results, the ethanolic extract of *G. malaccensis* (CC_50_ > 80 µg/mL) showed potentially less toxicity than the *M. siamensis* (CC_50_ < 5 µg/mL) and *M. ferrea* (CC_50_ = 11.44 µg/mL) extracts. SI value is a crucial parameter for determining whether further works can be continued for the extracts [20]. High SI indicates the potential antimalarial activity and safer therapy, whereas low SI indicates that the antimalarial activity is probably due to cytotoxicity rather than activity [21]. Among the three extracts that exhibited good antimalarial activity, *G. malaccensis* exhibited the highest SI > 15.01. Therefore, the ethanolic extract of *G. malaccensis* had the ability to protect against *P. falciparum* infection, albeit with safety concerns in mammalian cells. Therefore, this extract was further selected to confirm the in vitro antimalarial activity and toxicity in a mouse model using a 4-day suppressive test and acute toxicity test, respectively. Our study demonstrated that the ethanolic extract of *G. malaccensis* exhibited a significant dose-dependent reduction in parasitemia in *P. berghei*-infected mice. Unfortunately, the acute toxicity results showed that all mice died after a single dose of 2000 mg/kg. Regarding the Kheaw Hom remedy, the results from the in vitro study indicate that this remedy exhibited moderate antimalarial activity against *P. falciparum*. It was not considered to further study in a mouse model. Therefore, the result from this research is insufficient evidence to support the usage of this folk remedy to treat malaria.

The studied medicinal plant extracts of the Kheaw Hom remedy showed a high diversity of phytoconstituents, consisting of flavonoids, terpenoids, alkaloids, tannins, anthraquinones, saponins, and coumarins. Previous studies on the antimalarial activities of African medicinal plants have found that compounds in African flora, including alkaloids, terpenoids, flavonoids, and coumarins, which were also found in our extracted plants, showed promising in vitro antimalarial activities against various strains of *P. falciparum* [22,23,24,25,26,27]. This may be because these compounds contain one or more phenolic OH group each, leading to easy dimerization [27]. Numerous free OH groups might have anti-inflammatory activity, such as the ability to inhibit NO production and scavenge free radicals, such as DPPH [10,11]. These factors may lead to an increase in antimalarial activity.

The ethanolic extract of *G. malaccensis* (family Zingiberaceae) contained a diverse set of phytoconstituents, including flavonoids, terpenoids, alkaloids, saponins, and coumarins. In the antiplasmodial and cytotoxicity assays, the extract of this plant had the ability to protect against the K1 strain of *P. falciparum* with less toxicity to mammalian cells. Previous studies on *G. malaccensis* found that the chemical constituents in the rhizome of *G. malaccensis*, which were divided into two groups by structure (sesquiterpenoids (isocurcumenol, curcumenol, and zedoarondiol) and curcumenoid (curcumin)), had the ability to inhibit cAMP phosphodiesterase with moderate to high activity [28,29]. In malaria, cyclic nucleotide signaling from both cAMP and cGMP plays a crucial role in the development and differentiation of organisms, acting as second messenger molecules of malaria parasites [30,31,32,33]. Therefore, the ethanolic extract of *G. malaccensis,* which contains terpenoids, might affect the development and differentiation of *Plasmodium* parasites, leading to a decrease in parasitemia. Remarkably, flavonoids, which are abundant plant phenolic compounds, were moderately present in the ethanolic extract of *G. malaccensis* and may be the active compounds in this plant that play a crucial role in inhibiting parasite growth, because flavonoids have been reported to inhibit the intraerythrocytic growth of chloroquine-sensitive and chloroquine-resistant strains of *P. falciparum* [34]. Flavonoids are believed to act as antimalarials by inhibiting the biosynthesis of fatty acids in the parasite. They also probably act by inhibiting the influx of L-glutamine and myoinositol into infected erythrocytes during the intraerythrocytic phase of the *Plasmodium* life cycle [35]. In addition, flavonoids have antimalarial effects due to their antioxidant capacity, which is from their redox properties. Flavonoids can prevent the increase in free radical production during malaria infection in both the host and parasite under severe oxidative stress [36].

Based on the results of the in vitro antimalarial activity, our study demonstrated that the ethanolic extract of *M. siamensis* flowers possessed the most potent antimalarial activity. The phytochemical screening revealed the presence of flavonoids, terpenoids, tannins, and coumarins. Previous phytochemical investigation showed that the flowers of *M. siamensis* contain several geranylated coumarins such as mammeasin A, B, C, and D, together with mammea coumarins such as mammea E/BA cyclo D, mammea E/BC cyclo D, mammea E/BD cyclo D, and mammea E/AC cyclo D [37,38]. Therefore, coumarins probably act as the active compound of *M. siamensis* that exhibits antimalarial effects. Coumarin compounds isolated from the root of *Angelica gigas* demonstrated significant inhibitory activity against chloroquine-sensitive *P. falciparum* strains [39]. However, the exact mechanism of action of coumarins against *P. falciparum* has not yet been reported. 

In addition, the phytochemical analysis of *M. ferrea* flowers showed that this extract contains flavonoids, terpenoids, tannins, anthraquinones, saponins, and coumarins. Coumarins and terpenoids are the major groups of compounds that have been isolated and characterized from the flower of *M. ferrea*. Some coumarins include 5,7-dihydroxy-4-(1-hydroxypropyl)-8-(2-methylbutanoyl)-6-[3,7-dimethylocta-2,6-dienyl]-2H-chromen-2-one and 5,7-dihydroxy-8-(2-methylbutanoyl)-6-[3,7-dimethylocta-2,6-dienyl]-4-phenyl-2H-chromen-2-one, and some terpenoids include trans-caryophyllene, α-humulene, γ-muurolol, β-selinene, germacrene D, and β-bisabolene [40]. For terpenoids, it might interfere with the parasite’s polyisoprenoid biosynthesis through the inhibition of the isoprenyl diphosphate synthases, which condense molecules of isopentenyl pyrophosphate (IPP) among other isoprenic substrates to form isoprene chains [41]. In addition, the possible mode of action of anthraquinones as antiplasmodial agents is vaguely understood. Nonetheless, anthraquinones may inhibit *P. falciparum* growth by inducing oxidative stress [42]. 

The 4-day suppressive and acute toxicity tests in malaria-infected mice revealed a decrease in parasitemia after treatment with all doses of the *G. malaccensis* extract, but all mice died after receiving a high dose of the *G. malaccensis* extract. This may be caused by phytotoxins and heavy metals such as lead (Pb) and cadmium (Cd) that are found in the plant family Zingiberaceae [43,44].

Finally, some limitations of this study should be noted. First, we used acute toxicity tests to investigate the toxicity of the *G. malaccensis* extract in an animal model. The results showed that all of the mice died after receiving the high dose of *G. malaccensis* extract. This led to our study lacking data about the biochemical changes and histological observations of the livers and kidneys of the mice. Therefore, future experiments should select up-and-down procedures for acute toxicity. Additionally, this study did not investigate the active compounds from the Kheaw Hom remedy and its medicinal plant ingredients that possess high antimalarial activity. Therefore, further isolation and identification of the active compounds, the mechanism of action, and toxicity profile are required to evaluate its antimalarial potential.

## 5. Conclusions

The ethanolic extract of *G. malaccensis* rhizomes, a Thai medicinal plant of the Kheaw Hom remedy, revealed potent antimalarial activity against the *P. falciparum* K1 strain and *P*. *berghei*, which could serve as a promising starting point for antimalarial drugs. Isolation of the compounds and identification of specific chemical constituents from *G. malaccensis* as well as safety evaluations seem to be of special interest for further antimalarial studies. The Kheaw Hom remedy possessed moderate activity. Consequently, further work should be performed to characterize the active compounds and study their pharmacokinetics and toxicity. 

## Figures and Tables

**Figure 1 tropicalmed-07-00368-f001:**
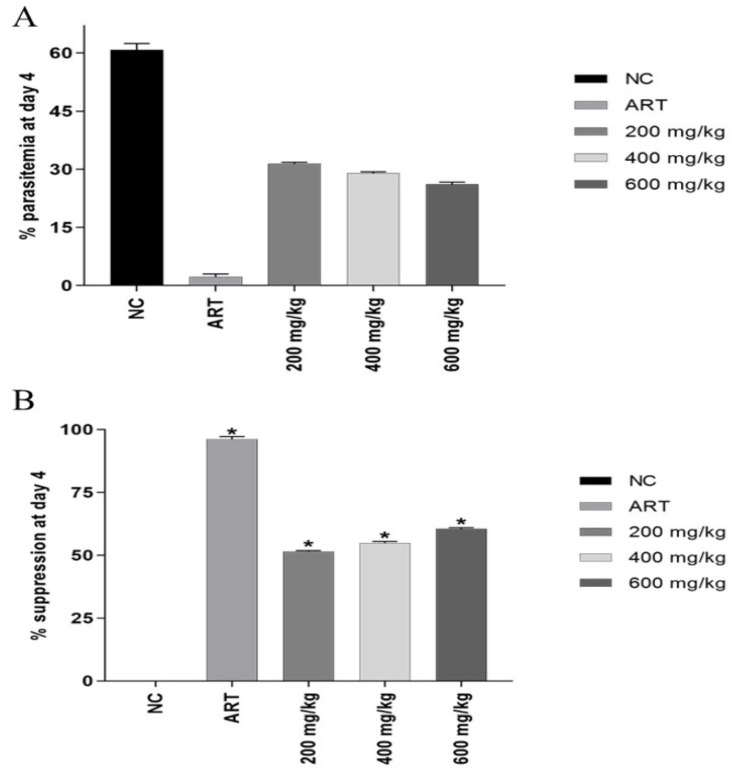
Effect of ethanolic extract of *G. malaccensis* (200, 400, and 600 mg/kg body weight) on the percent parasitemia (**A**) and percent suppression (**B**) in *Plasmodium berghei*-infected mice in the 4-day suppressive test compared to the negative control (NC) and positive control (artesunate; ART). Data are expressed as mean ± SEM (n = 5 per group), * Compared to negative control, *p* < 0.05.

**Table 1 tropicalmed-07-00368-t001:** List of plant materials and parts used in the Kheaw Hom remedy.

No	Plant Species	Family	Plant Part	Voucher Number
1	*P. cablin* (Blanco) Benth.	Lamiaceae	Leaves	SMD 142031002
2	*L. rugosa* (Roth) Merr.	Scrophulariaceae	Leaves	SMD 211012017
3	*C. fruticosa* (L.) A.Chev.	Asparagaceae	Red leaves	SMD 004002001
4	*C. fruticosa* (L.) A.Chev.	Asparagaceae	Green leaves	SMD 004002002
5	*E. stoechadosmum* Hance.	Asteraceae	Leaves	SMD 072036002
6	*V. zizanioides* (L.) Nash ex Small	Gramineae	Roots	SMD 119094001
7	*K. galanga* L.	Zingiberaceae	Rhizomes	SMD 288013007
8	*T. hoaensis* Pit.	Rubiaceae	Stem	SMD 233076005
9	*D. loureiroi* Gagnep.	Dracaenaceae	Stem	SMD 096001007
10	*A. evecta* (G. Forst.) Hoffm.	Marattiaceae	Rhizomes	SMD 165001001
11	*G. malaccensis* Ridl.	Zingiberaceae	Rhizomes	SMD 288010008
12	*T. chantrieri* André.	Taccaceae	Rhizomes	SMD 262001001
13	*S. exigua* Craib.	Leguminosae-Papilionoideae	Roots	SMD 148070001
14	*C. podophylla* (Hook.) Copel.	Cyatheaceae	Stem	SMD 084001007
15	*M. elengi* L.	Sapotaceae	Flowers	SMD 042004001
16	*M. ferrea* L.	Guttiferae	Flowers	SMD 122007001
17	*M. siamensis* Kosterm.	Guttiferae	Flowers	SMD 122006002
18	*N. nucifera* Gaertn.	Nelumbonaceae	Stamens	SMD 181001001

**Table 2 tropicalmed-07-00368-t002:** Group classifications and doses used in the 4-day suppressive test.

Groups (n = 5/group)	Extract/Drug	Dose (mg/kg)
Negative control	DMSO	-
Positive control	Artesunate	6
Experimental group 1	*G. malaccensis* ethanolic extract	200
Experimental group 2	*G. malaccensis* ethanolic extract	400
Experimental group 3	*G. malaccensis* ethanolic extract	600

**Table 3 tropicalmed-07-00368-t003:** Phytochemical screening of ethanolic and aqueous extracts of the medicinal plants in the Kheaw Hom remedy.

Plant	Extract	Phytochemical Constituents							
		FL	TN	AL	TA	AN	CG	SA	CM
*P. cablin*	Ethanolic	+	-	++	++	-	-	-	-
	Aqueous	-	-	++	++	-	-	+	-
*L. rugosa*	Ethanolic	-	+	-	+	-	-	-	+
	Aqueous	-	-	-	+	-	-	-	-
*C. fruticosa*	Ethanolic	+	-	++	+	-	-	-	-
(Red leaves)	Aqueous	-	-	-	+	-	-	++	-
*C. fruticosa*	Ethanolic	+	-	++	+	-	-	-	-
(Green leaves)	Aqueous	-	-	-	+	-	-	++	-
*E. stoechadosmum*	Ethanolic	-	+	-	+	-	-	-	-
	Aqueous	-	-	+	+	-	-	-	-
*V. zizanioides*	Ethanolic	+	+	-	-	-	-	-	+
	Aqueous	+	-	++	+	-	-	+	-
*K. galanga*	Ethanolic	-	+	-	-	-	-	-	-
	Aqueous	+	-	+++	-	-	-	+	-
*T. hoaensis*	Ethanolic	-	+	-	+	+	-	-	-
	Aqueous	+	-	++	+	-	-	+	-
*D. loureiroi*	Ethanolic	++	+	-	-	-	-	-	-
	Aqueous	++	+	-	+	-	-	-	+
*A. evecta*	Ethanolic	+	-	-	+	+	-	+	-
	Aqueous	+	-	+	+	+	-	-	-
*G. malaccensis*	Ethanolic	++	+	-	-	-	-	+	-
	Aqueous	+	+	+	-	-	-	++	+
*T. chantrieri*	Ethanolic	+	+	-	-	-	-	-	-
	Aqueous	+	-	+	+	-	-	++	-
*S. exigua*	Ethanolic	+++	+	-	-	-	-	++	+
	Aqueous	+	-	+++	-	-	-	-	-
*C. podophylla*	Ethanolic	+	+	-	++	-	-	+	-
	Aqueous	+	-	-	+++	-	-	-	+
*M. elengi*	Ethanolic	++	+	-	+	-	-	-	-
	Aqueous	++	-	-	+	-	-	++	-
*M. ferrea*	Ethanolic	+	+	-	++	+	-	++	+
	Aqueous	+	-	-	++	-	-	+	-
*M. siamensis*	Ethanolic	++	+	-	+	-	-	-	+
	Aqueous	+	-	-	+	+	-	+	+
*N. nucifera*	Ethanolic	+	+	-	-	-	-	-	-
	Aqueous	+	-	-	+	+	-	+	+

FL: flavonoids; TN: terpenoids; AL: alkaloids; TA: tannins; AN: anthraquinones; CG: cardiac glycosides; SA: saponins; CM: coumarins. +++: highly present; ++: moderately present; +: low presence; -: absent.

**Table 4 tropicalmed-07-00368-t004:** In vitro antimalarial activity and cytotoxicity of the crude extracts from the Kheaw Hom remedy.

Plant	Ethanolic Extract			Aqueous Extract		
	IC_50_ (µg/mL)	CC_50_ (µg/mL)	SI	IC_50_ (µg/mL)	CC_50_ (µg/mL)	SI
*P. cablin*	24.49 ± 0.01	>80	>3.27	549.30 ± 0.07	>80	>6.87
*L. rugosa*	54.02 ± 3.73	>80	>1.48	83.76 ± 1.58	>80	>1.05
*C. fruticosa* (Red leaves)	71.26 ± 2.40	>80	>1.12	245.00 ± 13.18	>80	>3.06
*C. fruticosa* (Green leaves)	68.31 ± 1.85	>80	>1.17	245.00 ± 13.18	>80	>3.06
*E. stoechadosmum*	28.15 ± 0.73	>80	>2.84	634.60 ± 32.60	>80	>7.93
*V. zizanioides*	158.70 ± 17.7	>80	>0.50	546.30 ± 13.20	>80	>6.83
*K. galanga*	25.80 ± 0.63	>80	>3.10	630.30 ± 14.05	>80	>7.88
*T. hoaensis*	72.02 ± 0.88	>80	>1.11	397.20 ± 13.20	>80	>4.97
*D. loureiroi*	10.47 ± 0.01	55.67 ± 3.80	5.32	104.00 ± 0.01	>80	>1.30
*A. evecta*	57.57 ± 2.89	>80	>1.39	105.10 ± 3.94	>80	>1.31
*G. malaccensis*	5.33 ± 0.08	>80	>15.01	103.10 ± 11.23	>80	>1.29
*T. chantrieri*	65.45 ± 3.54	>80	>1.22	80.63 ± 2.51	>80	>1.01
*S. exigua*	224.40 ± 3.82	18.56 ± 0.56	0.08	173.20 ± 25.81	>80	>2.17
*C. podophylla*	312.50 ± 5.05	>80	>0.26	329.40 ± 5.05	>80	>4.12
*M. elengi*	49.80 ± 2.44	>80	>1.61	106.20 ± 7.63	>80	>1.33
*M. ferrea*	7.39 ± 0.89	11.44 ± 1.14	1.55	112.20 ± 8.05	>80	>1.40
*M. siamensis*	1.50 ± 0.03	<5	<3.33	82.29 ± 1.54	>80	>1.03
*N. nucifera*	157.90 ± 8.61	>80	>0.51	53.08 ± 1.43	>80	>0.66
Kheaw Hom remedy	14.16 ± 0.65	>80	>5.65	292.90 ± 15.15	>80	>3.66
Artesunate	1.28 ± 0.71 *		ND	ND		ND
Doxorubicin	ND		ND	2.07 ± 0.13		ND

SI: Selectivity index. Data are presented as the mean ± SEM. ND = not determined. * IC_50_ unit expressed as ng/mL.

**Table 5 tropicalmed-07-00368-t005:** Effects of the ethanolic extracts of *G. malaccensis* on the percent parasitemia in *Plasmodium berghei*-infected mice in the 4-day suppressive test.

Group	Dose (mg/kg)	% Parasitemia	% Suppression
Negative control	-	64.22 ± 0.27	-
Artesunate	6	2.56 ± 0.83	96.02 ± 1.35 ^a,c,d,e^
*G. malaccensis* extract (Treated groups)	200	31.10 ± 0.21	51.57 ± 0.36 ^a,b,e^
	400	28.94 ± 12.94	54.94 ± 0.58 ^a,b,e^
	600	25.34 ± 11.33	60.53 ± 0.47 ^a,b,c,d^

Data are presented as the mean ± SEM (n = 5 per group). ^a^ Compared to negative control, ^b^ compared to artesunate, ^c^ compared to 200 mg/kg extract, ^d^ compared to 400 mg/kg extract, ^e^ compared to 600 mg/kg extract, *p* < 0.05.

**Table 6 tropicalmed-07-00368-t006:** Effects of the ethanolic *G. malaccensis* extracts on body weight changes in *Plasmodium berghei*-infected mice in the 4-day suppressive test.

Group	Dose (mg/kg)	Day 0 (g)	Day 4 (g)	Change (%)
Negative control	-	36.56 ± 0.56	35.47 ± 0.49	−3.82 ± 0.33 ^b,c,d,e^
Artesunate	6	35.95 ± 0.90	37.04 ± 0.97	6.42 ± 0.72 ^a,c,d,e^
*G. malaccensis* extract (Treated groups)	200	35.19 ± 0.29	34.56 ± 0.34	−1.78 ± 0.68 ^a,b,d,e^
	400	35.08 ± 0.23	32.58 ± 0.36	−7.11 ± 0.67 ^a,b,c,e^
	600	34.60 ± 1.15	30.37 ± 1.04	−12.21 ± 0.72 ^a,b,c,d^

Data are presented as the mean ± SEM (n = 5 per group). ^a^ Compared to negative control, ^b^ compared to artesunate, ^c^ compared to 200 mg/kg extract, ^d^ compared to 400 mg/kg extract, ^e^ compared to 600 mg/kg extract, *p* < 0.05.

## Data Availability

The data of this study are available from the corresponding author on request.

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
