# Peer review of "Exploring Potential Antimalarial Candidate from Medicinal Plants of Kheaw Hom Remedy"

_tropicalmed, 2022, doi:10.3390/tropicalmed7110368_

Round 1
Reviewer 1 Report
11- Error: 2.4. In Vitro Cultivation and Maintenance of Plasmodium Falciparum. The specie must be falciparum.
22- What commercial kit was used for cytotoxicity in the MTT colorimetric assay?
33- In 2.7. Four-Day Suppressive Test (Peter’s Test). Why the negative control group received 200 μl of 7% Tween 80 solution? The correct solution will be water or ethanol.
4- How many animals per group were placed in the experiment? In the tables are n=5. Is this correct?
55- The authors should show the parasitaemia curves of the mice infected with P. berghei to better discuss the effects of the extracts, compared to the drug used as a cure control (artesunate). Because, they said: “The 4-day suppressive and acute toxicity tests in malaria-infected mice revealed a decrease in parasitemia after treatment with all doses of the G. malaccensis extract, but all mice died after receiving a high dose of the G. malaccensis extract “.
66- If the authors made histological sections of the kidneys, they should show and discuss these results to relate it to the cytotoxicity tests.
77- What is the conclusion regarding the statistical tests?
Author Response
"Please see the attachment."

Reviewer 2 Report
This is a well written manuscript that clearly presents the evaluation of Kheaw Hom remedy as an antimalarial, correctly concluding that no further work should be done on it to treat malaria. The authors systematically work through the 18 constituent plants to test ethanolic and aqueous extracts on K1 and subsequent cytotoxicity. They prioritise one extract on the basis of it's potency and lack of cytotoxicity in Vero cells. This is dosed in vivo to show modest suppression at high dose; at a dose 3 times higher than this, the mice died within 48h.
Unless I missed it, it would be helpful to have the dose of Kheaw Hom used for treating fever mentioned in the introduction to help frame the question being asked.
The key questions that I have for the authors are as follows:
1. Although phytochemical analysis was performed and there was extensive inferred discussion about the agents responsible for potency and toxicity, what is actually known about the specific chemical compound components of each plant extract?
2. Why did the authors not fine fractionate to narrow down and isolate the precise component leading to the antimalarial activity? As it stands, it appears that a complex mixture has been tested on the parasite and in cytotoxicity assays (and in vivo). This means the precise identity of the active component(s) are unknown and thus their antimalarial and cytotoxicity activities.
3. The in vivo studies were well argued and the triaging to select the extract to test was sound. Clearly, all ethical considerations were adhered to. However, on what basis were the doses selected for the efficacy experiment? Was any kind of pharmacokinetic analysis also performed? I assume not given the absence of knowledge of ingredients, but it is important for the reader to understand. In essence - the in vivo study gives limited information but, in it's current form, is unhelpful in really understanding the concotion's potential.
4. I don't understand why the 2g/kg acute toxicity study was performed in mice. All that was learnt was that the mice died. Given an absence of knowledge of the active or toxic ingredients (which may or may not be the same) no comment on therapeutic margins or reasons for toxicity can be given with certainty.
5. I would alter language around "promising candidate" to "promising starting point" as a complex mixture having potency on K1 of 1.5ug/ml is far from being candidate quality. The authors should explain that to take this forward, given that the Kheaw Hom remedy is not suitable for malaria (a good conclusion of the paper) isolation of the active components and careful characterisation would need to be performed to understand the chemistry, the intrinsic potency, toxicity, pharmacokinetics etc.
6. Most of the discussion was fairly vague given the absence of clarity of the active ingredients. The authors have, in fairness, reported on biochemistry of known natural product families from these plants but it is far from being a rigorous study - thus it almost would be better to simply inform the reader that more work is needed to understand the mode of action and toxicity and potential for malaria.
Author Response
"Please see the attachment."

Reviewer 3 Report
Punsawad and coworkers describe the antiplasmodial/antimalarial evaluation of Kheaw Hom remedy extracts on P. falciparum K1 (CQ-resistant) and P. berghei infected mice. In its current form, the work may be acceptable for publication in TropicalMed. However, the reviewer has a few queries:
1. What about the resistance indices of G. malaccensis’ ethanolic and aqueous extracts?
2. It would be interesting to determine the active components of M. siamensis, which, while cytotoxic, shows activity comparable to Artesunate.
Author Response
"Please see the attachment."

Round 2
Reviewer 2 Report
Thank you for responding to the comments.
Author Response
"Please see the attachment." i
